# Singularity response reveals entrainment properties in mammalian circadian clock

Kosaku Masuda [1,2], Naohiro Kon[3,4,5,6], Kosuke Iizuka[3,4], Yoshitaka Fukada [5,7], Takeshi Sakurai [1,2] ✉ & Arisa Hirano [1,2] ✉

Entrainment is characterized by phase response curves (PRCs), which provide a summary of responses to perturbations at each circadian phase. The synchronization of mammalian circadian clocks is accomplished through the receipt of a variety of inputs from both internal and external time cues. A comprehensive comparison of PRCs for various stimuli in each tissue is required. Herein, we demonstrate that PRCs in mammalian cells can be characterized using a recently developed estimation method based on singularity response (SR), which represents the response of desynchronized cellular clocks. We confirmed that PRCs can be reconstructed using single SR measurements and quantified response properties for various stimuli in several cell lines. SR analysis reveals that the phase and amplitude after resetting are distinguishable among stimuli. SRs in tissue slice cultures reveal tissue-specific entrainment properties. These results demonstrate that SRs can be employed to unveil entrainment mechanisms with diverse stimuli in multiscale mammalian clocks.

The circadian clock plays a crucial role in enabling many organisms to adapt to diurnal environmental changes. In mammals, circadian clocks have been shown to exhibit a strong correlation with various physiological processes, and thus, the misalignment of the circadian rhythm could increase the risk of diseases such as depression, cancer, and diabetes[1–3]. Notably, recent studies have demonstrated that the alteration of entrainment of the circadian rhythm, such as chronic jet lag as a model of shift work or time-restricted feeding, has a significant impact on metabolism in mice and humans[4–7]. Consequently, elucidation of the entrainment properties of the circadian clocks is crucial for improving human health and extending lifespan[8].

The primary circadian pacemaker is located in the suprachiasmatic nucleus (SCN), while nearly all peripheral tissues possess their own peripheral clocks, and various internal and external cues are involved in their synchronization[9,10]. In vivo peripheral clocks are entrained by the nervous as well as the endocrine systems; for instance, the rhythmic secretory pattern of glucocorticoids, which is controlled by the SCN, plays a role in synchronizing peripheral clocks[11]. Rhythmic changes in body temperature serve as resetting signals to broadly coordinate the circadian rhythms throughout the body[12,13]. In addition to internal signals, the light-dark cycle, food intake, and medical interventions also affect the phase of circadian rhythms[14,15]. To fully understand how these diverse stimuli contribute to alterations in circadian rhythms at a systemic level, a comprehensive study of their entrainment properties is necessary. However, most studies have focused on investigating the effects of individual stimuli on phase resetting of cellular clocks, and the differences and interactions among these stimuli and tissues have yet to be fully explored. Furthermore, as

[1]Institute of Medicine, University of Tsukuba, Tsukuba, Ibaraki 305-8577, Japan. [2]International Institute for Integrative Sleep Medicine (WPI-IIIS), University of Tsukuba, Tsukuba, Ibaraki 305-8577, Japan. [3]Institute of Transformative Bio-Molecules (WPI-ITbM), Nagoya University, Furo-cho, Chikusa-ku, Nagoya 464-8601, Japan. [4]Laboratory of Animal Integrative Physiology, Graduate School of Bioagricultural Sciences, Nagoya University, Furo-cho, Chikusa-ku, Nagoya 464-8601, Japan. [5]Department of Biological Sciences, Graduate School of Science, The University of Tokyo, Hongo, Bunkyo-ku, Tokyo 113-0032, Japan. [6]Suntory Rising Stars Encouragement Program in Life Sciences (SunRiSE), 8-1-1 Seikadai, Seika-cho, Soraku-gun, Kyoto 619-0284, Japan. [7]Laboratory of Animal Resources, Center for Disease Biology and Integrative Medicine, Graduate School of Medicine, The University of Tokyo, Hongo, Bunkyo-ku, Tokyo 113-0033, Japan. ✉e-mail: sakurai.takeshi.gf@u.tsukuba.ac.jp; hirano.arisa.gt@u.tsukuba.ac.jp

each peripheral clock in different tissues shows a unique responsiveness to entrainment cues, a simple and versatile method that is applicable to a wide range of circadian systems (from single cell to tissue levels) is needed.

The dynamics of the entrainment of circadian rhythms with external stimuli are traditionally determined using phase response curves (PRCs)[16]. Circadian rhythms exhibit phase responses (i.e., phase advance or delay) to stimuli depending on the phase of the rhythm when the stimulus is given. A PRC represents a summary of the phase response to the stimulus at each phase (Supplementary Fig. 1a). The amplitude of the PRC represents the entrainable range, which is the range of period length within which the circadian rhythm can be entrained. The phase to which the circadian rhythm is entrained with a periodic stimulus is represented as the phase of the stable point of the PRC, at which the phase response changes from positive to negative. The PRC is usually obtained by applying a stimulus at several phases and measuring the resultant phase shift (advance or delay). This is an expensive and time-consuming process, as it requires replicated experiments at different phases (at least four time points, but usually more) across a day (Supplementary Fig. 1a).

Recently, we proposed a method to mathematically estimate PRCs in a single experiment using phase and amplitude responses at an extremely low amplitude state, known as a singularity state, in the plant circadian clock in *Arabidopsis*[17]. In the singularity state, circadian rhythms at the population level are apparently damped due to desynchronization within cells[18]. When a stimulus is applied to cells in the singularity state, circadian rhythms usually reset their phase with specific amplitudes at the population level, depending on the stimulus and its strength (Supplementary Fig. 1b). This is known as the rhythm resetting "singularity response," abbreviated as SR, reflecting the primal characteristics of the PRC for that stimulus. Essentially, when the amplitude of PRC is large, a large SR is observed (Supplementary Fig. 1c). The phase of the stable point of PRC corresponds to the phase of the SR (Supplementary Fig. 1d). Therefore, SR allows for the efficient estimation of PRC through a simple single experiment, as opposed to conventional methods (Supplementary Fig. 1a, b). Furthermore, since PRC parameters can be obtained for each sample using SR, differences in responsiveness can be statistically evaluated. However, the usefulness of SR has not yet been evaluated in mammalian circadian clocks.

In this study, we validated the theoretical background[19] for the SR-based PRC estimation method (termed SR method) in mammalian cell culture using data obtained in a previous study, in which the authors had determined the PRC through the imaging of cellular rhythms and their responses to stimuli at the single-cell level[20]. We established the efficiency of the SR method by comparing the parameters of the PRC and SR at the single-cell level. Additionally, we demonstrate the versatility of the SR method by showing its applicability to various cell lines and its utility in quantitively determining the entrainment properties of various stimuli. Furthermore, we measured SRs in tissue slice cultures and determined the tissue-specificity of the entrainment of the mouse circadian clock.

## Results

### Reciprocal relationship between phase response and amplitude in single-cellular and population rhythms

First, we verified the theoretical underpinnings of the SR method, which was predicated on the relationship between the amplitude of circadian rhythms and PRC[16,19,21]. Typically, phase responses of circadian rhythms at lower amplitude states are stronger than those at higher amplitude states. However, the phase of the stable point, wherein the phase response shifts from positive to negative, remains unchanged. The amplitude of circadian rhythms can be attributed to two primary factors: the amplitude at the cellular level, and synchronization rate of the cell population (Fig. 1a). The amplitude dependency of PRC at the single-cell level is explained as a decrease in

amplitude with a relative increase in the intensity of the stimulus, which enhances the phase response[16,21], as depicted in Fig. 1b. In contrast, the amplitude dependency of PRC at the population level is a result of desynchronization in the cell population[19,22,23]. When cells are highly synchronized, the phase response of the population approximates that of individual cells at the same phase. Conversely, when cells are desynchronized, the response of the population rhythm is the averaged response of all desynchronized cells. In this state, the average of each phase response remains constant, irrespective of the phase of the population rhythm. In other words, the population rhythm consistently resets to a specific phase and amplitude as shown in Fig. 1b. Therefore, phase changes become larger as the population becomes desynchronized. The variations in the PRCs, which depend on amplitude of the cellular and population rhythms, were found to be quite similar (Fig. 1b; detailed in Supplementary Note 1)[19,24]. In both models, the PRC shifts from a type-1 PRC (continuous) to a type-0 PRC (discontinuous) as the rhythm amplitude decreases. However, the phase of the stable point is unaltered by the amplitude in both cases. Therefore, the rhythm in the extremely low amplitude state is always reset to a phase of the stable point after stimulation. Figure 1c shows the amplitude response curve, which describes the amplitude responses to the perturbation at each phase. An amplitude response basically shows an increase at the stable point of the PRC and a decrease at the unstable point, where the phase response changes from negative to positive. At singularity, the amplitude is reset to a consistent value irrespective of phase in both models. Therefore, the circadian rhythm is reset to a certain phase and amplitude by a stimulus at the singularity irrespective of whether it was caused by damping of the cellular rhythms or desynchronization of the cellular rhythms in a cell population.

We verified the amplitude dependencies of PRCs at both the single-cell and population levels using experimental data. Here, we used data from a previous study, in which the PRC was measured through single-cell imaging of *Reverbα-Venus-NLS-PEST* reporter in NIH3T3 cells[20]. We calculated the amplitude of *Reverbα-Venus* fluorescence rhythms before and after the stimulation and divided these into two groups according to amplitude values. Figure 1d shows the PRCs of the cells with high or low amplitudes for forskolin, phorbol-12-myristate-13-acetate (PMA), and sodium hydrogen bicarbonate (NaHCO₃). Although it was difficult to distinguish between changes in the shape of PRCs due to a large amount of noise, the variation in phase responses increased around the unstable points as the amplitude of the cellular rhythm decreased, while it did not change significantly around the stable point (Supplementary Fig. 2). We then analyzed the rhythms of the cell population. We established random small populations containing five individual cells and calculated the synchronization rate among these five rhythms and the phase of the population rhythm. The phase response of the cell population also showed a change from type-1 to type-0 PRC as amplitude decreased (Fig. 1e). The distribution of stable points was not significantly different between the high and low amplitude states, as observed at the single-cell level (Supplementary Fig. 3). The same results were observed in larger populations ($N = 10$ or $20$ cells, Supplementary Fig. 4a). However, the larger populations showed a bias in the phase distribution and a decrease in samples categorized as synchronized populations (Supplementary Fig. 4a). Thus, we also established populations containing cells with similar phases, which were intentionally selected to obtain highly synchronized populations (Supplementary Figure 4b). In both cases, the phase responses of the cell populations changed from type-1 to type-0 PRC as amplitude decreased, as shown in the model. Amplitude responses were also observed at the single-cell level and the population level, where amplitude increased around the stable time point and decreased around the unstable point, as predicted by the models (Supplementary Fig. 5). These results confirmed that both models reset to a constant phase and amplitude in extremely low

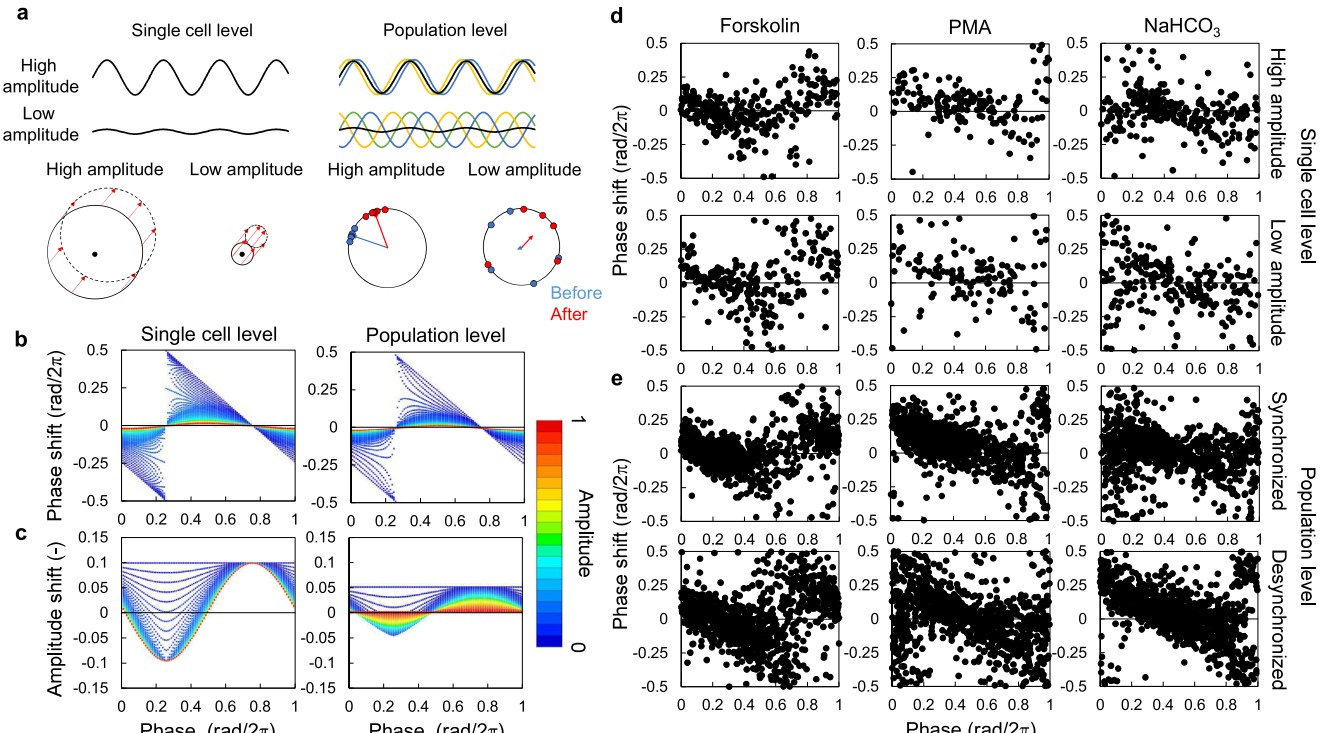

**Fig. 1 | Amplitude dependency of PRCs at the cellular and population levels.**
**a** Models of circadian rhythm and phase response at the cellular and population levels. **b** Mathematical models for PRC at the single cell and population levels. **c** Mathematical models for amplitude response curve at the single cell and population levels. PRCs for forskolin, PMA, and NaHCO$_3$ at the single cell (**d**) and population levels (**e**). At the population level, one sample contains five cells, which are randomly selected. Concentrations of forskolin, PMA, and NaHCO$_3$ were 0.75 μM, 2 μM, and 77.75 mM for single-cell PRCs and 0.5 μM, 1 μM and 66.5 mM for population PRCs, respectively. In single-cellular PRCs, an amplitude of $A > 0.6$ was defined as high amplitude, while $A < 0.4$ was considered low amplitude. In population PRCs, an amplitude (synchronization index) of $R > 0.8$ was defined as synchronized, while $R < 0.1$ was considered desynchronized.

amplitude states, depending on the PRC. Therefore, we concluded that the singularity response can be observed whether the singularity state is caused by damped cellular rhythm or desynchronization.

## Validation of the SR method in NIH3T3 cells

A previous study[20] demonstrated that even when the amplitude at the population level is extremely low, about half of the cells were rhythmic at the single-cellular level in NIH3T3 cells (Supplementary Fig. 6). In other words, the singularity state is a result of desynchronization in cultured cells rather than a decrease in the rhythm amplitude in a single cell. Therefore, to estimate PRCs in this study, we assumed that the singularity is basically caused by the desynchronization of the cell population for mathematical modeling. In the previous study, PRCs at the single-cell level were obtained through imaging individual cellular rhythms, which were desynchronized within cells[20]. Figure 2a shows the phase response of individual cells to forskolin in NIH3T3 cells. The phases of the cells were disparate before the stimulus but converged to a certain phase after it. We then averaged single-cellular rhythms to examine the response of the population rhythms. Because the cellular rhythms were desynchronized, the amplitude of the averaged rhythms was extremely low and the response to the stimulation at the population level could be considered an SR, as we proposed previously. As expected, the population rhythms (average of individual rhythms) in the singularity state were reset due to the stimulus, indicating that SR could be observed in the population of NIH3T3 cells (Fig. 2a). We compared the PRC obtained through individual cellular responses and SR for a series of concentrations of forskolin (Fig. 2b, c). The amplitude of the PRC monotonically increased with its increasing concentration, while the amplitude of SR tended to increase with its concentration except for the highest concentrations (Fig. 2d). The PRC and SR showed almost similar values of phase when the stimulus was strong

enough (Fig. 2e). We also analyzed SRs to other resetting agents, including NaHCO$_3$, PMA, LiCl, and CoCl$_2$ (Fig. 2f). The population rhythms were reset after these stimuli, and the reset phase and amplitude of each response depended on the chemicals. We compared the parameters calculated using the PRC and SR for each condition of the treatment (Fig. 2g, h). The amplitudes of the PRC and SR were significantly proportional. The phases in the PRC and SR were almost similar at many points, while large errors were observed at some time points presumably because the stimulus was too weak (Supplementary Fig. 7).

We used the previously proposed method of estimating PRCs using SR[17]. In our model, the amplitude of SR increases monotonically with the strength of input, and the amplitude of PRC increases as well (Supplementary Fig. 8a, b). The phase of SR also corresponds to the phase of stable point of the PRC (Supplementary Fig. 8c, d). Therefore, PRC can be estimated according to the phase and amplitude of SR. In fact, we confirmed a similarity between estimated PRCs (solid red lines in the figure) and measured PRCs (solid black circles), which were determined using single-cellular responses (Fig. 3). The stable time points of both PRCs were consistent for each stimulus. The amplitudes of both PRCs were also similar and significantly larger than that in the untreated condition, indicating that the responsiveness could be very well characterized using the SR method. Furthermore, the change in the shape of PRCs depending on the strength of treatments could also be recapitulated using the SR method, as demonstrated by the PRC in a condition of 6 μM forskolin (Fig. 3).

## Validation of the SR method in Rat-1 cells

We also verified the SR method in Rat-1 cells (Fig. 4). We used Rat-1 cells stably expressing *Bmal1-luc* reporter[25] and treated them with TGF-beta, melatonin, PMA, forskolin, and dexamethasone (DEX). Populations of

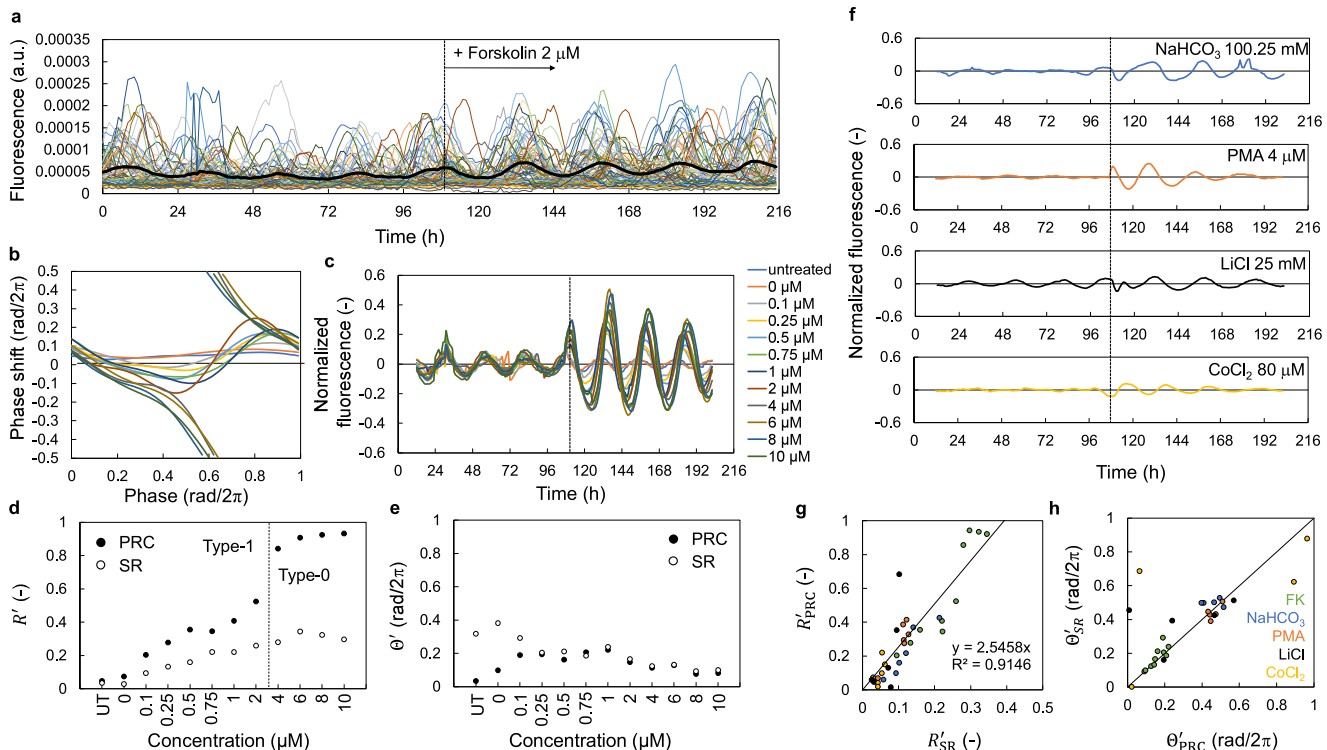

**Fig. 2 | Relationship between parameters of PRC and SR in NIH3T3 cells.**
**a** Fluorescence signals derived from *Reverbα-Venus-NLS-PEST* reporter at single-cellular level are shown. The black line is averaged fluorescence signal, indicating the population rhythms. A stimulus of 2 μM forskolin was applied to cells 108 h after the beginning of the measurement. **b** PRCs for forskolin treatment at different concentrations at single-cellular level. **c** SRs to forskolin at different concentrations. Fluorescence rhythms of all individual cells were averaged and then normalized. The dotted line indicates the treatment of forskolin. **d** Amplitude parameter *R'* in

SRs and PRCs. "UT" represents untreated condition. **e** Phase parameter Θ' in SRs and PRCs. **f** SRs to indicated resetting agents. Fluorescence rhythms of all individual cells were averaged and then normalized. The dotted lines indicate the treatments. **g** Amplitude of PRCs and SRs for different concentrations of agents. The linear black line is the regression line for calibration parameter $\beta$ in Eq. (7). $R^2$ is the coefficient of determination. FK means forskolin. **h** Phase of PRCs and SRs for different concentrations of agents. The linear black line indicates the line of $y = x$.

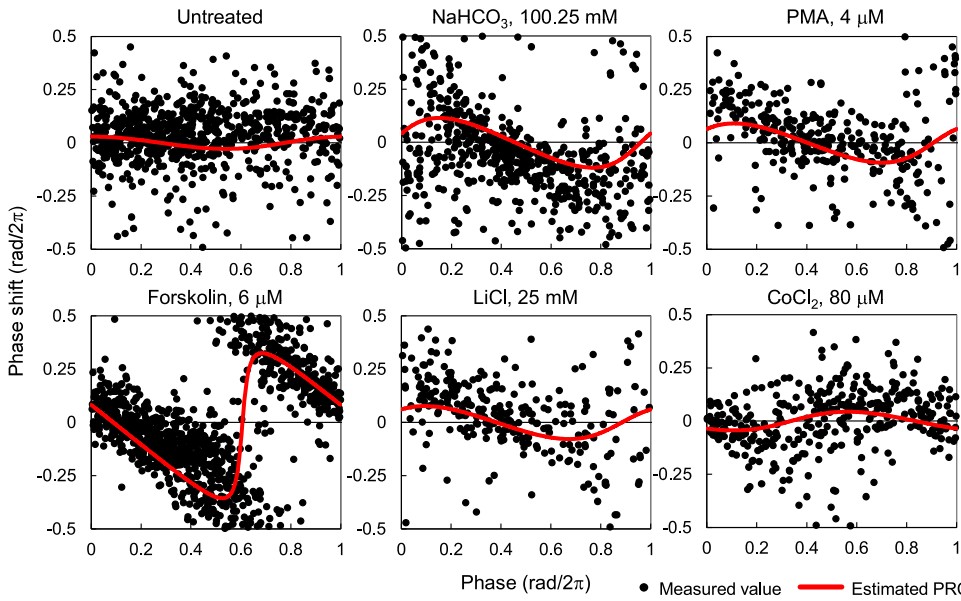

**Fig. 3 | Estimation of PRC using SR parameters.** Black points indicate the phase responses of individual cells to indicated chemicals, which were measured at the single-cell level. Red lines indicate the estimated PRCs using the SR parameters.

cultured cells were desynchronized over time due to environmental disturbances or variations in the circadian period of each cell as observed in NIH3T3 cells (Fig. 2a). In fact, the population level rhythm of the Rat-1 cells decayed over time and reached the singularity state at

day 5 after recording initiation (Fig. 4a), when stimulus was applied to induce SRs. The phase and amplitude of SRs for each stimulus are shown in Fig. 4b. As expected, treatment with DEX, which is known to elicit a strong phase reset of the circadian rhythms[11], induced the

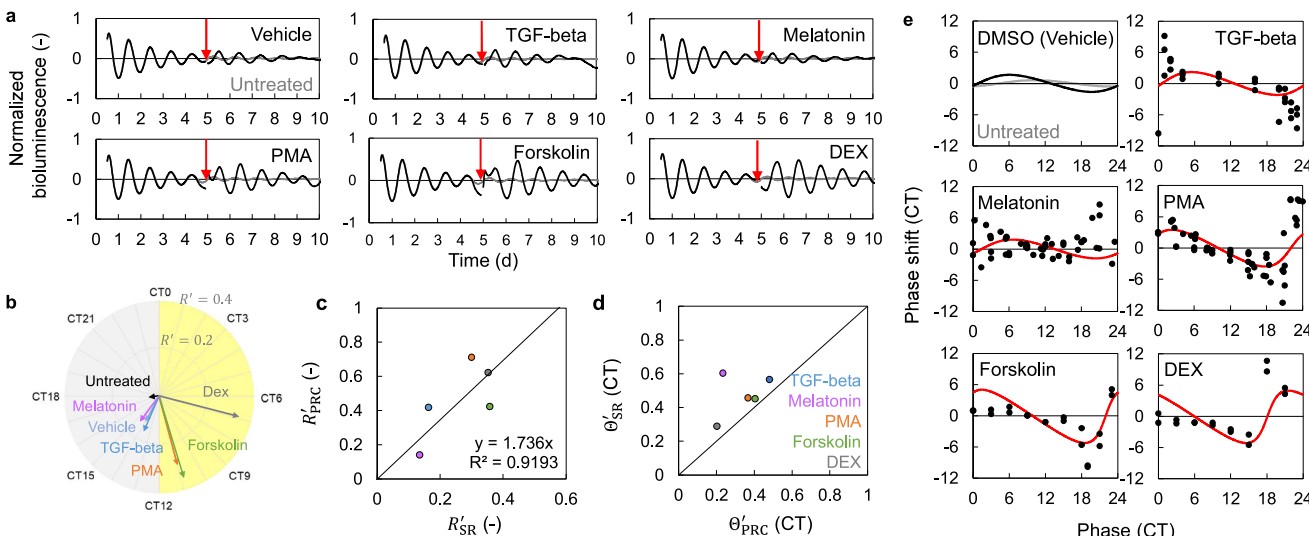

**Fig. 4 | PRC estimation in Rat-1 cells. a** Normalized bioluminescence in Rat-1 *Bmal1-luc* cells (Mean, n = 8 biologically independent samples (Untreated), 6 (Vehicle), 10 (others)) showing SRs to various stimuli. Cells were treated with vehicles or indicated chemicals (2 ng/ml TGF-beta, 20 nM melatonin, 10 μM PMA, 10 μM forskolin or 100 nM DEX). Black and gray lines indicate the rhythms w/wo stimulation, respectively. Red arrows indicate the time of the stimulation. **b** Amplitude and phase of SRs. The distance from the center point represents largest response (Supplementary Fig. 9a) and the phase of resetting varied depending on stimulus (Supplementary Fig. 9b). To confirm accuracy of the SR parameters, the parameters of PRCs were also measured using the conventional method, wherein stimuli were applied to cells at several circadian phases across a day to determine PRCs (Fig. 4c, d). Amplitudes of SRs were generally proportional with those of PRCs, and phases of SRs were consistent with those of PRCs, except for those observed after melatonin treatment, which induced the weakest response (Fig. 4c, d). The estimation of PRC was also successful for each treatment (Fig. 4e). These results indicate that the SR method can be applied to different cell lines and with different clock gene reporters, provided the stimulus is strong enough.

amplitude, and the angle represents the phase. The peak time of *Bmal1-luc* activity was defined as CT0. **c** Amplitude of PRCs and SRs. The linear black line is the regression line for calibration parameter $\beta$ in Eq. (7). $R^2$ is the coefficient of determination. **d** Phase of PRCs and SRs. The linear black line indicates the line of $y = x$. **e** Measured PRCs and estimated PRCs. Points indicate the measured phase responses and solid curves indicate the estimated PRCs.

temperature changes and low-dose DEX treatment, were close to sine functions (Supplementary Fig. 11). On the other hand, those for strong stimuli, such as $H_2O_2$ and high-dose DEX treatments, were close to type-0 PRCs.

Next, we compared the SRs to a series of concentrations of DEX stimuli. The responsiveness showed a considerable change between 4 nM and 0.8 nM (Fig. 5d). Similarly, the SR amplitude exhibited very little increase in the low-dose condition (0–0.16 nM), while it exhibited a large increase in the higher dose condition (0.8 and 4 nM), reaching saturation in the condition of 20 nM (Fig. 5e). On the other hand, the resetting phases advanced gradually as the concentration increased and advanced significantly between 20 and 100 nM concentrations (Fig. 5f). In fact, the similar trend in phase response to DEX was observed when the phase response was measured using the conventional PRC measurement method (Supplementary Fig. 12). The phase of measured PRC also showed an advance in a higher dose of DEX treatment. These results indicate that the stable point of PRC may change at higher concentrations, even when the amplitude response was saturated, confirming that the dose-response of cellular circadian rhythms can be easily evaluated using SR. When the concentration was sufficiently high, the rhythm was reset to the same phase as SR regardless of the phase before stimulation, indicating that PRC can be well predicted using the SR methods.

In the conventional method of PRC measurement, DEX stimulation is commonly used to synchronize the phase among cells before the measurement. This kind of operation is necessary because the change in the synchronization rate among cells alters the phase response at the population level (Fig. 1). Therefore, it is difficult to verify the effect of prior DEX treatment on the phase response to other stimuli in the conventional method. On the other hand, in the SR method, the response of the circadian rhythm is measured in a desynchronized state; therefore, it does not require synchronization of the clock through DEX treatment as in the conventional method. Here, we evaluated the effects of synchronization with DEX on the responses to temperature stimuli. At the beginning of the measurement, the amplitude of the bioluminescence rhythm was slightly higher in cells with the prior DEX treatment than that in untreated cells, but there was

## SRs for various stimuli in *PER2::LUC* MEF

To demonstrate the usefulness of SRs, we measured SRs for various kinds of stimulation in mouse embryonic fibroblasts (MEFs) prepared from *PER2::LUC* knock-in mice[9]. The normalized luminescence of *PER2::LUC* in cells treated with temperature stimuli, medium exchange, DEX, and oxidative stress ($H_2O_2$) are shown in Fig. 5a. Amplitudes of the rhythms decreased during the recording due to desynchronization and approached the singularity state before the introduction of stimulus. The rhythm resets upon application of stimuli at the singularity were observed, and the reset amplitude and phase were dependent on the stimulus. DEX treatment induced the strongest response, i.e., most of the cells were reset to the same phase (Fig. 5b), as shown in Fig. 4. Phase responses to the temperature stimuli were smaller than the other treatments but significantly larger than those in the untreated condition (Fig. 5b). Each stimulus induced a reset to a different phase (Fig. 5c and Supplementary Fig. 10). In particular, the positive and negative temperature perturbations induced phase resets in almost opposite directions. In previous studies, PRCs for DEX and $H_2O_2$ indicate the stable time point at CT8 and CT14, respectively[26,27], which is consistent with our results (Fig. 5c). The temperature cycle also entrained the rhythms so that the trough of *PER2::LUC* rhythm (approximately CT0) shifted to the peak of the temperature cycle[28]. These results indicate that the phase of PRC can be efficiently estimated using SRs in MEFs. The estimated PRCs for weak stimuli, such as

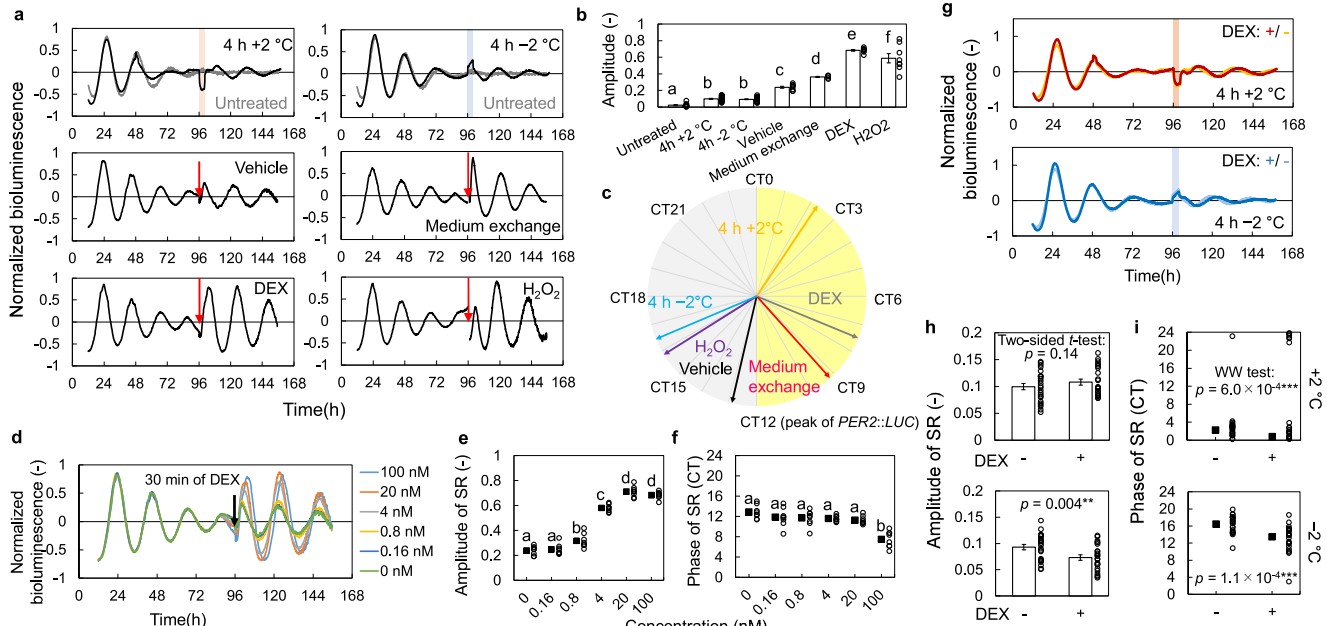

**Fig. 5 | Evaluation of phase response properties using SRs in MEF *PER2::LUC*.**
**a** SRs to indicated stimulations. Red arrows and colored region indicate time of stimulation. The concentration of dexamethasone (DEX) and H$_2$O$_2$ were 100 nM and 2 mM, respectively. **b** Amplitude of SRs (*n* = 5 biologically independent samples in medium exchange, 24 in temperature stimulation and 8 in the others, Mean ± SEM). Each two conditions with different letters represent significant differences (Tukey–Kramer test, *p* < 0.05). **c** Phase of SRs (the peak of *PER2::LUC* was defined as CT12). **d** SRs to DEX treatment for 30 min at various concentrations. Dose effects of

DEX on amplitude (**e**) and phase (**f**) of SRs (*n* = 8 biologically independent samples, mean ± SEM in **e** and circular mean in **f**). Each two conditions with different letters are significantly different (Tukey–Kramer test in **e** and Watson–Williams test (WW test) with Bonferroni correction in **f**, *p* < 0.05). **g** Normalized luminescence during the SR measurement for the temperature stimuli with and without prior DEX treatment. Amplitude (**h**) and phase (**i**) of SR for temperature stimuli with/without prior synchronization using DEX (*n* = 24 biologically independent samples, mean ± SEM in **h** and circular mean in **i**).

no significant difference upon application of the temperature stimulus in both conditions (Fig. 5g). On the other hand, the amplitude of SR to cooling stimulus was slightly weakened in the DEX-treated cells (Fig. 5h). The phases of SR to both heating and cooling stimuli were advanced by the prior DEX treatment (Fig. 5i). These results indicate that synchronization of cellular rhythms by DEX may alter the stable point of PRC for temperature stimulation. Therefore, the SR method likely represents the phase responses closer to the natural characteristics than does the conventional method which uses DEX for synchronization.

## Measurement of SRs in slice cultures

We have demonstrated that SR is useful to characterize the entrainment properties of the circadian clock in cultured cells. However, the characteristics of circadian rhythms in individual animals depend on organs and tissues[9,29]. In terms of entrainment of circadian rhythms, the SCN is responsible for light entrainment, while the peripheral clocks seldom respond to light but are entrained to other signals such as temperature or feeding cycles[10,14,15]. However, measuring tissue-specific PRCs is much more tedious than it is for cultured cells. Here, we examined whether SR can also be recorded in tissue slice cultures. We measured SRs to temperature stimuli (+2 °C for 4 h) in slices of the liver, kidney, lung, white fat, spleen, and muscle prepared from *PER2::LUC* mice. The rhythms in each tissue decayed with time, although more slowly than in cultured cells, and reached the singularity state on day 9 in all tissues (Fig. 6a). The amplitude at the beginning of the measurement showed tissue specificities (Fig. 6b), while no significant differences were found in the circadian periods (Fig. 6c). We then compared the parameters of SRs when the rhythms were reset in response to temperature stimulus (+2 °C for 4 h). The rhythms after the stimulation showed larger amplitude than that before the stimulation at least in lung and kidney (Fig. 6d). The phases of SRs in slice cultures were at around CT12 (Fig. 6e and Supplementary

Fig. 13). This result is consistent with those of previous studies, since the stable point of PRC for positive temperature stimulation in lung was around CT15 in a previous study[13]. On the other hand, MEFs showed an SR phase around CT2, which is almost opposite to the phase of the lung slice in culture. This suggests that the synchronization characteristics change depending on the tissue and cell type. While comparing the SR phases and the circadian periods of slice culture, a negative correlation was observed (Supplementary Fig. 14). In general, the longer the period the greater would be the delay in locking phase. Therefore, this result suggests that each tissue maintains a constant phase relationship under the temperature cycles as the difference in PRC phase cancels out the shift in locking phase due to the difference in period.

The present work demonstrated that the SR method is a versatile method that can be used to easily and quantitatively evaluate characteristics of the phase response in the mammalian circadian clocks from cellular to tissue levels.

## Discussion

Herein, we propose a simple PRC estimation method using SRs in mouse and rat circadian clocks. First, we analyzed the single-cell PRC data measured in NIH3T3 cells[20] and verified the theoretical background for the SR method. We confirmed the amplitude dependencies of PRC at both single-cellular and population levels. Furthermore, we demonstrated that SR can be used to estimate PRC for NIH3T3 and Rat-1 cells. Since SR measurement is quite simple, it can easily evaluate the dose effect of stimulation on phase response, and phase response to stimulation with drugs. In addition, we succeeded in measuring SRs at the tissue level using slice cultures and clarified the tissue-specificity of entrainment properties. Our results indicate that PRC estimation using SR is an effective method for elucidating the entrainment properties of the circadian clock, and that the mouse circadian clock exhibits diverse entrainment properties.

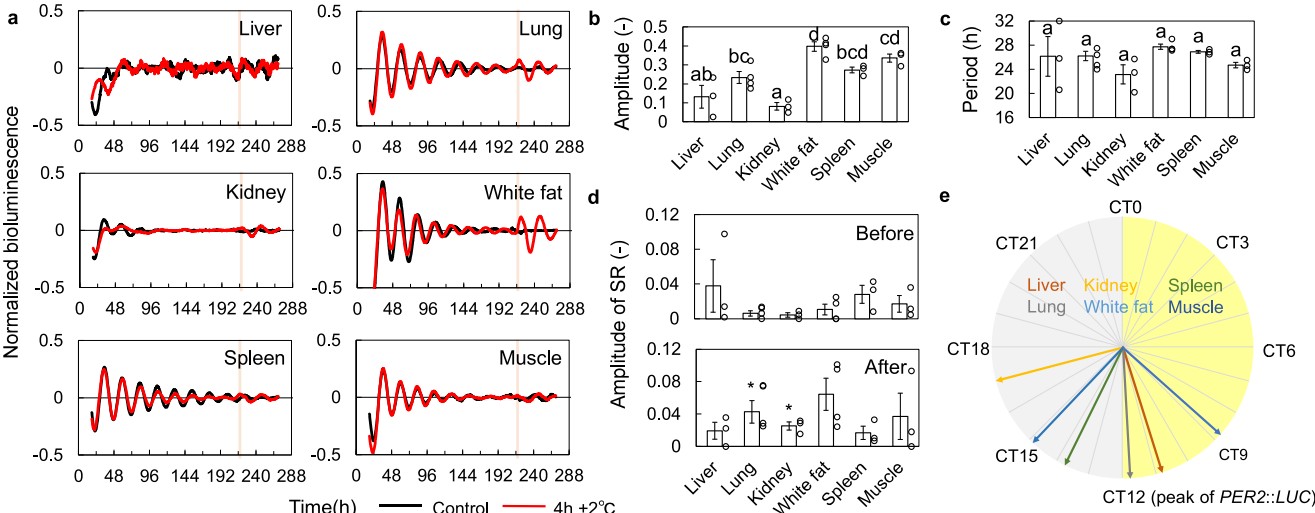

**Fig. 6 | SRs for heating stimulation in slice cultures. a** SRs of different tissues to 4 h + 2 °C stimulation. Red and black lines indicate the rhythms with and without stimulation, respectively. Vertical bars indicate temperature stimuli. Amplitude (**b**) and period (**c**) of each tissue (mean ± SEM, $n = 4$ independent experiments for lung and white fat and 3 for the others). Each two condition with different letters represents significant differences (Tukey–Kramer test, $p < 0.05$). **d** The amplitude before and after the stimulation (mean ± SEM, $n = 4$ independent experiments for lung and white fat, $n = 3$ independent experiments for the others). Asterisks indicate significant differences (t-test, $p < 0.05$). **e** Phase of SRs in different tissues (circular mean).

In this study, we showed that changes in rhythm amplitude at the cellular and at the population levels alter the phase responses, and that the responses at the very low amplitude state are similar in both cases (Fig. 1). However, since about half of the cells were actually rhythmic in NIH3T3 cells[20], we assumed that the singularity (low amplitude) state was generally caused by desynchronization between cells, and not by dampened single-cellular rhythms (Fig. 2a and Supplementary Fig. 7). Under this assumption, we succeeded in estimating PRCs from the SR parameters measured in both NIH3T3 and Rat-1 cells. However, the amplitude of the rhythm at the cellular level may be attenuated with some treatment. Therefore, here we discuss the relationship between PRC and SR with decay of amplitude at single-cell level. First, we assume that the phase response at the single-cell level is expressed in a simple limit-cycle model (Eqs. (S1) and (S2) in Supplementary Note 1). When the amplitude is zero in all cells, the amplitude and phase after stimulation are expressed by the intensity of stimulus $F$ and the direction $\varphi$. Based on these parameters, we can estimate the change in PRC depending on the amplitude of the rhythm. The change in PRC when the rhythm amplitude is lower is similar to that of the cell-population model (Fig. 1b), and the difference in amplitude of these PRCs can be calibrated by comparing the SR amplitude with the measured PRC amplitude as shown in Fig. 2g. Therefore, even if we assume the decay of the single-cellular amplitude, the estimated PRC may be similar to that estimated using the cell-population model. On the other hand, the amplitude at the single-cell level and the synchronization rate of the cell population may change simultaneously. However, even if we consider the population of the limit-cycle oscillators, the amplitude dependency of PRC would not be changed from the case of single cell as shown in Eq. (S11). This result suggests that, at least for rough estimation of PRC, there is no need to make a strict distinction between the rhythm amplitude at the single-cell level and synchronization rate of the population for determinant of the amplitude of population rhythms. In other words, whether the singularity state is caused by damping of single-cellular amplitude or desynchronization, the amplitude and phase of SR reflect the PRC amplitude and phase, respectively.

We measured SRs using two different cell lines and reporter genes: Rat-1 *Bmal1-luc* and *PER2::LUC* MEF. They showed similar phase responses to DEX treatment. Therefore, these cell lines were thought to have similar entrainment properties for DEX. It is noteworthy that the reset phases for various stimuli such as negative temperature, hormones, cytokines, and oxidative stress were concentrated between CT6 and CT18 (Figs. 4b and 5c). In NIH3T3 *Reverbα-Venus* cells in the study by Manella et al.[20], there was no stimulus that could reset the phase of circadian rhythm to 0.6–0.8 (rad/2π), which is opposite to the reset phase against forskolin (Fig. 2h). Another previous study also showed that the reset phases for various treatments such as cAMP, epidermal growth factor (EGF), and basic fibroblast growth factor (bFGF) were similar to each other[30]. These results suggest a common mechanism of the phase response for various stimuli. However, in terms of medication for circadian rhythm disorders, circadian rhythms should be advanced or delayed according to the patient's symptoms. Thus, drugs that exhibit distinct reset features/mechanisms are needed. In our study, the positive temperature stimulus exceptionally exhibited a reset phase close to CT0 (Fig. 5c), suggesting its unique property of phase response. Therefore, elucidation of the mechanisms by which temperature stimuli induce phase responses may be one target for drug development for circadian rhythm control.

We showed that SR is an effective way to estimate the synchronization properties of circadian clocks easily. However, PRCs may have complicated shapes such as the dead zone, where the circadian clocks show almost no responses as shown in Fig. 4e. The SR method proposed in this study cannot estimate PRCs with such a complex shape. Therefore, this PRC estimation method may be more effective for use in the first high-throughput screening such as drug discovery for circadian rhythm regulation, where a vast amount of screening including dose effect and organ-dependency is required. In fact, the SR parameter is useful for the quantification and statistical analysis of dose effects (Fig. 5). A recent study developed another high-throughput PRC measurement method using dual reporter genes that enable cell tracking and measurement of circadian rhythms at the single-cell level (Circa-SCOPE[20]). This method uses the principle that in the desynchronized state, the responses at different phases can be considered at once, like in the SR method. In this method, the exact phase responses are measured; therefore, the shapes of PRCs are also accurately evaluated, unlike in the SR method. However, because this method measures the rhythm of individual cells through imaging, it requires continuous measurement over a certain period of time, a large amount of data, and high cost for analysis. In contrast, our proposed SR method requires only the population rhythm after the stimulation.

Therefore, it is possible to achieve both accuracy and efficiency by assessing responsiveness using the SR method first and then obtaining accurate PRCs using the single-cell tracking method.

In the mammalian circadian clock system, the master clock resides in the SCN of the hypothalamus. However, we did not perform SR measurements on the SCN slice cultures in this study, because the SCN has stronger cell-to-cell coupling than the other organs, making spontaneous desynchronization difficult. However, tetrodotoxin (TTX), which is a Na⁺ channel blocker, inhibits the coupling among neurons in the SCN, causing desynchronization of circadian rhythms[31], and a previous study showed SR-like responses to temperature stimulation under TTX treatment[13]. Therefore, TTX treatment may enable the PRC estimation using SR even in the SCN, although we need to consider the possibility that TTX suppresses rhythmicity or changes the response characteristics of circadian rhythms at single-cell level and/or that the intercellular coupling may also affect the phase response properties[32]. Therefore, the entrainment properties in the SCN and the effect of intercellular coupling on phase response, including application of the SR method, may need to be examined deeply.

## Methods

### Animals
All experiments involving animals were approved by the Animal Experiment and Use Committee of the University of Tsukuba and were therefore in accordance with NIH guidelines. Animals were maintained under a 12:12 h light/dark cycle at $23.5 \pm 1.0\,°C$ and $50.0 \pm 10.0\%$ humidity. Food and water were available *ad libitum*.

### Bioluminescence analysis using Rat-1 fibroblasts
We used Rat-1 *Bmal1-luc* cells generated in the previous study[25]. Real-time monitoring of cellular circadian gene expression in Rat-1 cells was performed as described previously[25]. Briefly, Rat-1 cells expressing *Bmal1*-luc reporter were plated on 35-mm dishes ($1.0 \times 10^6$ cells/dish) and cultured at $37\,°C$ under 5% $CO_2$ in phenol-red-free DMEM (catalog no. D2902, Merck KGaA, Darmstadt, Germany) supplemented with 10% fetal bovine serum (FBS, Biowest, Pays de la loire, France), 3.5 mg/ml D-glucose, 3.7 mg/ml NaHCO₃, 50 U/ml penicillin and 50 μg/ml streptomycin. To measure PRCs, 24 h after the plating, the cells were treated with 100 nM DEX for 2 h, and the medium was replaced with a recording medium [DMEM supplemented with 10% FBS, 3.5 mg/ml D-glucose, 25 U/ml penicillin, 25 μg/ml streptomycin, 0.1 mM D-Luciferin potassium salt (catalog no. 126-05116, FUJIFILM Wako Pure Chemical, Osaka, Japan or catalog no. E1601, Promega) and 10 mM HEPES-NaOH; pH 7.0]. The bioluminescence signals from the cells were continually recorded at $37\,°C$ under air with dish-type bioluminescence detector LumiCycle 32 Color (Actimetrics, Wilmette, IL, USA). Cells were treated with DEX, PMA, melatonin, TGF-beta, or forskolin 118 h after the start of recording. To measure SRs, the bioluminescence was recorded without DEX pre-treatment. Five days after the start of recording, SR was induced by treatment with DEX, PMA, melatonin, TGF-beta, or forskolin. Concentrations presented in the text are final concentrations.

### Bioluminescence analysis using *PER2::LUC* MEF
We used *PER2::LUC* MEFs established from *PER2::LUC* knock-in mice generated in the previous study[9]. MEFs were cultivated in high-glucose DMEM with L-glutamine, phenol red, and sodium pyruvate (catalog no. 043-30085, FUJIFILM Wako Pure Chemical, Osaka, Japan) supplied with 10% FBS (Merck KGaA, Darmstadt, Germany), 100 U/ml penicillin and 100 μg/ml streptomycin (catalog no. 09367-34, NACALAI TESQUE, Kyoto, Japan) at $37\,°C$ under 5% $CO_2$. Bioluminescence recording medium was phenol-red-free DMEM (catalog no. D2902, Merck KGaA, Darmstadt, Germany) with 3.5 mg/ml D-glucose and 10 mM HEPES adjusted to pH 7.0, and it was supplemented with 10% FBS, 100 U/mL

penicillin, 100 μg/ml streptomycin, and 0.1 mM D-luciferin potassium salt luciferin (catalog no. 126-05116, FUJIFILM Wako Pure Chemical, Osaka, Japan) just before use. MEFs were plated in 24-well plates with 500 μl of recording medium 1 day before the measurement. Bioluminescence was monitored for 1 week at intervals of 10 min using an automated monitoring device, Kronos HT (ATTO, Tokyo, Japan). In each experiment, stimuli were applied 96 h after the start of the measurement, and responses were measured over the following 3 days. In the temperature stimulation, the temperature was controlled by the monitoring device, and heating ($39\,°C$) and cooling ($35\,°C$) stimuli from $37\,°C$ were given for a duration of 4 h. For pre-synchronization in temperature stimulus conditions, 100 nM of DEX was applied 30 min before the start of measurement. In chemical stimulations, first, half of the culture medium was removed to another plate, and then 25 μl of each agent diluted in PBS was added. The medium was replaced 30 min later with the removed medium. Concentrations presented in the text are final concentrations. Measurements of SR for each stimulus in MEFs were performed twice. We also measured PRC for DEX stimulation by the traditional PRC measurement method. To measure the responses of high amplitude rhythms, we stimulated the cells 30, 36, 42, or 48 after the beginning of measurements without prior DEX treatment. We gave the DEX stimulation in the same manner as the SR measurement, and three samples were measured for each concentration and timing of stimulus. We obtained the phase only after the stimulation by cosine fitting because the data before the stimulation were too short to be normalized and cosine fitted. Measurements of PRC for each DEX stimulus in MEFs were performed once.

### Bioluminescence analysis using slice culture
For measurement of tissue slice culture, we used heterozygous male *PER2::LUC* knock-in mice (C57BL/6J genetic background, 31-39 weeks old). Tissue slices were cut out from mice at ZT6 to ZT7, and each tissue piece was placed on a cell culture insert (catalog no. PICM01250, Merck KGaA, Darmstadt, Germany) in a 24-well plate with 500 μl of the recording medium (phenol-red-free DMEM (Merck KGaA, Darmstadt, Germany) with B-27 supplement (Thermo Fisher Scientific, Waltham, MA, US), 3.5 mg/ml D-glucose, 10 mM HEPES, 35 mg/l NaHCO₃, 100 U/ml penicillin, 100 μg/ml streptomycin and 0.2 mM D-luciferin potassium salt, pH 7.0). Monitoring was performed using Kronos HT, and heating stimulus (4 h, $39\,°C$) was applied 9 days ($n = 3$ replications in lung and white fat, 2 in others) or 11 days ($n = 1$ in each tissue) after the beginning of the measurement. In each experiment, two or three tissue slices from one mouse were measured, and the one that showed the strongest luminescence after the stimulation was used for analysis.

### Analysis for PRC at single-cell level
For analysis of amplitude dependency of PRCs, we used the single-cellular PRC measurement data using NIH3T3 cells obtained in the previous study[20]. As in the previous study, phase and amplitude were obtained using cosine fitting for normalized bioluminescence before (24–96 h) and after (120–164 h) the stimulation. Normalization was performed as follows,

$$L_j = \frac{l_j - \bar{l}_j}{\bar{l}_j}, \tag{1}$$

$$\bar{l}_j = \frac{1}{2n+1} \sum_{k=0}^{2n} l_{k+j-n}. \tag{2}$$

Here, $l_j$ is bioluminescence at the $j$th time point, $\bar{l}_j$ is a moving average with a 24 h window, $L_j$ is normalized bioluminescence and $n$ is the number of data points within 12 h ($n = 12$ in the case of NIH3T3 cells).

Cosine fitting was performed with the least square method using Python (lmfit; https://lmfit.github.io/lmfit-py/). The period of the fitting curve was limited to 16–32 h. The rhythms with coefficients of determination less than 0.5 were excluded from the calculation for PRC. We also defined amplitudes greater than 0.6 as high amplitude and those less than 0.4 as low amplitude.

## Analysis for PRC at population level

Five cells were randomly selected from the single-cell data (approximately 1000 cells in each condition), and the average of each rhythm was determined as the population rhythm. The synchronization rate $R$ of the five cell rhythms was calculated as per the following equation,

$$R = \left| \frac{1}{n} \sum_{j=1}^{n} e^{i\theta_j} \right|. \tag{3}$$

Here, $\theta_j$ is phase of each cell and $n$ = 3, 5, 10 or 20. Phase response was obtained in the same way as in the single-cell PRC. Repeating this calculation, the PRC at synchronized state (1000 points with synchronization rates $R > 0.8$) and at desynchronized state (1000 points with synchronization rates $R < 0.1$) were obtained. We also composed the synchronized population of the cells with similar phase. For each cell, we selected the cells that are closest in phase to it, then calculated the phase response of the population rhythm.

## SR method

The procedure for estimating PRC from SR parameters using the SR method followed that established in the previous study[17]. First, experimental data were normalized using Eqs. (1) and (2), and cosine fitting was performed on the data one day after the stimulation to define SR phase and amplitude.

Next, the SR parameters of PRC were defined using the following equation,

$$R'_{PRC} e^{i\Theta'_{PRC}} = \frac{1}{2\pi} \int_0^{2\pi} e^{i(\phi + g_{exp}(\phi) + \omega \triangle t)} d\phi. \tag{4}$$

Here, $g_{exp}(\phi)$ is a phase response curve which was obtained experimentally, $\omega$ is a free-running frequency and $\Delta t$ is time duration of the stimulus. $g_{exp}(\phi)$ is obtained from the PRC data, which contained pairs of phase $\phi_j$ and phase shift $\Delta \phi_j$, by maximizing the sum of $\cos\{g_{exp}(\phi_j) - \Delta \phi_j\}$. We assumed two types of PRC, type-1 and type-0 PRC. The PRCs have the assumed forms,

$$\text{Type} - 1\,\text{PRC} : g_{exp,type-1}(\phi) = c_1 + c_2 \sin(\phi - c_3) + c_4 \sin(2\phi - c_5), \tag{5}$$

$$\text{Type} - 0\,\text{PRC} : g_{exp,type-0}(\phi) = c_1 + c_2 \sin(\phi - c_3) + c_4 \sin(2\phi - c_5) - \phi. \tag{6}$$

Here, PRCs were set to the second order harmonics because there is some noise in the experimentally measured PRCs and it is difficult to accurately determine the higher order components of the PRC. Both types of $g_{exp}(\phi)$ were fitted to PRC data, and the one that has a larger sum of $\cos\{g_{exp}(\phi_j) - \Delta \phi_j\}$ was adopted.

The SR parameters in experiments were defined as the amplitude and phase at the end of the stimulation. These parameters were calculated from the rhythms 24–48 h after the stimulation in the same manner as the PRC calculation. In the tissue slice data, a moving average with a 12 h window was performed before the fitting of sine curves. The average of phase data (circular mean) was calculated as $\bar{\theta} = \arg\{\frac{1}{n} \sum_{j=1}^{n} e^{i\theta_j}\}$.

We assumed that the measured value of the SR amplitude $R'_{SR}$ and the SR parameter $R'_{PRC}$ of the PRC amplitude are represented by the following relationship,

$$R'_{PRC} = \beta R'_{SR}. \tag{7}$$

The proportional coefficient $\beta$ accounts for differences in amplitude at the single-cell level due to the differences in experimental condition and cell lines. For NIH3T3, $\beta_{3T3} = 2.28$ from Fig. 2g, and for Rat-1, $\beta_{Rat-1} = 1.48$ from Fig. 4c. For MEF *PER::LUC2*, the SR amplitude saturates at about $R'_{SR} \cong 0.75$ in Fig. 4e, so we can estimate that $\beta_{MEF} = 1/0.75 \cong 1.33$ (Supplementary Note 2). The measured value of the SR phase $\Theta'_{SR}$ and the SR parameter $\Theta'_{PRC}$ of the PRC shows similar values as shown in Figs. 2h and 4d. Thus, we assumed that

$$\Theta'_{PRC} \cong \Theta'_{SR}. \tag{8}$$

Although the parameters of SR and PRC correspond to each other, the phase of PRC is not determined by SR parameters alone. However, it is a general property of a limit-cycle oscillator model that PRC that is type-1 for weak stimulation approaches type-0 as stimulation becomes stronger, so it is expected that similar results can be obtained regardless of the model. Therefore, we used the simplest phase oscillator model for estimation of PRC shape. From the previous study, let PRC at the single-cell level for the stimulation with duration $\Delta t$, $g_{model}(\phi, \Delta t)$, be obtained by the following a phase oscillator model,

$$\frac{d\phi}{dt} = \omega + E(t)Z(\phi), \tag{9}$$

$$Z(\phi) = a\sin(\phi - b), \tag{10}$$

$$E(t) = 1, t \in [0, \Delta t], \tag{11}$$

$$g_{model}(\phi(0), \Delta t) = \phi(\Delta t) - \phi(0) - \omega \Delta t, \psi(0) \in [0, 2\pi). \tag{12}$$

Here, $Z(\phi)$ is a phase sensitive function and assumed to be a simple sine function. $E(t)$ is a stimulus indicator and $E(t) = 1$ under the stimulation and $E(t) = 0$ otherwise. We did not include coupling within cell population, because coupling is not so important, at least if we are considering instantaneous phase response to stimuli. Even if the coupling causes a slight amplitude change, accurate SR parameters can be obtained by making the correction of Eq. (7). In this case, $g_{model}(\phi, \Delta t)$ is analytically solved (Supplementary Note 3)[17]. From Eq. (4) and the solution of $g_{model}(\phi, \Delta t)$, the relationship between $(R'_{PRC}, \Theta'_{PRC})$ and $(a, b)$ can be calculated. The SR parameters $(R'_{PRC}, \Theta'_{PRC})$ can be estimated from the measured SR parameters $(R'_{SR}, \Theta'_{SR})$ using Eqs. (7) and (8). Therefore, the parameters $(a, b)$ can be estimated from $(R'_{SR}, \Theta'_{SR})$.

In practice, given experimental data for SR with $R'_{SR}$ and $\Theta'_{SR}$, the parameters $a$ and $b$ were inversely estimated step-by-step. First, $R'_{PRC}$ and $\Theta'_{PRC}$ were obtained from $R'_{SR}$ and $\Theta'_{SR}$ using Eqs. (7) and (8). Then, by minimizing a square error between the experimental $R'_{PRC}$ and the theoretical $R'_{PRC}$ using the generalized reduced gradient method, the parameter $a$ was estimated. Because $R'_{PRC}$ increases from 0 to 1 monotonously with $a$, $a$ is uniquely determined by $R'_{PRC}$ (Supplementary Fig. S9a). Next, the parameter $b$ was estimated to fit the theoretical $\Theta'_{PRC}$ to the experimental $\Theta'_{PRC}$. $b$ is also uniquely determined by $\Theta'_{PRC}$, which ranges from 0 to $2\pi$ rad (Supplementary Fig. S9c). The functional relationships between $R'$-to-$a$ and $\Theta'_{PRC}$-to-$b$ indicate that they are in a one-to-one correspondence, making the estimation procedure straightforward without any local minima. The parameters $(a, b)$ obtained from these calculations were substituted into Eq. (9) and the PRC was calculated to estimate the PRC for the stimulation. In Fig. 3, the stimulus length $\Delta t$ is not clear, so the calculation was performed with $\Delta t = 1$ (h).

## Statistical analysis

We performed statistical analysis using R software (version 4.1.1; https://www.r-project.org/). In the multiple comparison test, we used the R package "multcomp" (version 1.4.18) for Tukey–Kramer test, and "NSM3" (version 1.16) for Steel–Dwass test. We also used the R package "circular" (version 0.4.93) for the assessment of circular data.

## Reporting summary

Further information on research design is available in the Nature Portfolio Reporting Summary linked to this article.

## Data availability

All data needed to evaluate the conclusions of this study are presented in the manuscript and/or Supplementary Information. Additional relevant data and materials may be requested from the authors. Source data are provided with this paper.

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

## Acknowledgements

We are grateful to H. Yoshitane for providing the *PER2::LUC* MEFs. This study was partially supported by Grants-in-Aid for JSPS Fellows (No. 22J00270 to K.M.), for Scientific Research (22K15157 to K.M.) and for Specially Promoted Research (17H06096 to Y.F.) provided by the Japan Society for the Promotion of Science (JSPS), Moonshot Research Development Program (AGL03337 to A.H.) provided by Japan Agency for Medical Research and Development (AMED) and Research grant (A.H.) (The Naito Foundation, Japan).

## Author contributions

K.M. and A.H. designed the research. N.K. and K.I. performed the experiments using Rat-1 cells and K.M. and A.H. performed the experiments using MEFs and slice cultures. KM analyzed the data. K.M., Y.F., T.S. and A.H. wrote the manuscript. Y.F., T.S. and A.H. provided project supervision. All authors discussed the results and implications and commented on the manuscript.

## Competing interests

The authors declare no competing interests.
