## [Peer Review File · Nature Communications]

Singularity response reveals entrainment properties in mammalian circadian clockREVIEWER COMMENTS

Reviewer #1 (Remarks to the Author):

This manuscript illustrates a new method for measuring circadian clock phase response curves (PRC), which was previously demonstrated in plants (the first author of the current manuscript is also the first author in the previous paper, [17] of the references). The method is based on the Singularity Response (SR) and it consists in applying a stimulus to a de-synchronized cell population. From the phase and amplitude of the population after stimulation, a mathematical model is then applied to recover the PRC parameters. In principle, the SR method is less costly and of more practical application than the classical PRC method, but it doesn't recover as many details.

The current work tests and illustrates this method for mammalian (mouse) cells, using several cell lines (NIH3T3, MEF, Rat1), as well as tissue slices; several different stimulus according to the cell line; and different clock reporters, also according to cell lines (Rev-erb, Per, Bmal1). The results are very interesting and comprehensive taking into account the range of cells, stimulus, and reporters tested. In a general way, the new method is in a fair-to-good correspondence with the more classical PRC methods, but when it comes to closer comparison of detailed properties of the SR and PCR, further work and studies are needed. Throughout the manuscript, some points could be further developed:

-In the validation for NIH3T3 cells, the authors perform tests with "small populations of 5 cells". It is not clear to me what is the goal of this test. It seems to me that 5 elements is too small for a population, I would expect groups of 20-30 cells in order to obtain reasonable averages. And how do you choose these "cell populations" (are they near neighbors?). On the other hand, how do "populations of 5 cells" compare with the population sizes in the other experiments, in other cell lines? Is it reasonable to assume that these small population results can be extrapolated to "real" populations?

- The results for Rat-1 cells are not always clear to understand and/or explain. First, the phases computed through SR seem to always be underestimated (all dots above the $y=x$ in Fig 4d). Is this a specific problem for these type of cells?

Second, the PRC and SR parameters for Forskolin and PMA are very close to each other -- as indicated by the corresponding dots in Figs. 4c,d.

However, the reconstructed PRC curve for PMA fit the points very badly, in contrast to the Forskolin curve (Fig. 4e). Moreover, the PMA and Forskolin stimulation curves are similar in amplitudes (both high after stimulation), which seems to indicate that PMA should also provide good results, as was indeed the case in NIH3T3 cells. Can these divergences be explained in some way?

- The results for MEF cells do not include comparison with the classical PCR, but rather focus on the dependence of the phase response on the stimulus concentration, in particular for Dexamethasone (Fig. 5e,f). This is an interesting application of the SR method, where the SR phase and amplitude parameters are used to characterize phase response as a function of DEX concentration. Since the comparison is among SR parameters only, some meaningful information can be obtained (see also my next comment).

- Regarding the SR method itself, it can be shown that the SR phase parameter is equivalent to the phase parameter of the PCR method, i.e. $\theta(\text{PCR}) = \theta(\text{SR})$. This is, however, not true for the amplitude parameter, since a relation of the form $R(\text{PCR}) = \beta * R(\text{SR})$ needs to be computed for each cell type and the factor β is different from cell line to cell line, and probably different for each experiment. Therefore, the interpretation of the amplitude parameter for SR can be a problem when recovering PCR and would require further study.

Nevertheless, in the case of comparison among SR experiments only (cf. the previous comment), the values $R(\text{SR})$ retain their meaning and can be used to compare and characterize results from similar experiments, such as the dependence of phase response on the concentration of the stimulus (see Fig. 5e,f). Indeed, as the authors suggest in the Discussion, the SR method could be more suitable for a preliminary, and faster, phase response analysis in experiments such as high throughput screening, where large amounts of tests need to be performed. The SR would enable a selection of the best cases, which would then be followed by a more detailed classical PCR.

Other remarks:

Fig 1b,c: the captions need to be corrected, as the two panels include both single cell and population level(?)

Figs 2g and 4c: give a better idea of what the "approximate line" is, or at least refer to eq. (7) in the Methods.

Line 562 (Methods): Should be Fig 2g? (and not 2f)

Reviewer #2 (Remarks to the Author):

The authors apply their recently published (Masuda et al. Nat. Commun. 2021) singularity response (SR) method for inferring the phase response curve (PRC) from a single experiment to mammalian rather than plant circadian clocks. The method is successfully applied to experimental data in cultured mouse and rat fibroblast cells, as well as several mouse tissue slices for many different stimuli including temperature, PMA, DEX, Forskolin, sodium bicarbonate, LiCl, and CoCl₂. This variety of cell types and stimuli provide compelling evidence that the method is broadly useful. This is exciting since it promises to reveal the very insightful PRC with a single experiment from a desynchronized state rather than numerous experiments from a synchronized state for many different phases of the input.

After reading the methods section multiple times and consulting the previous paper on the SR method (Masuda et al. Nat. Commun. 2021), the SR method remains opaque. As I understand it, the method steps are 1) measurements of the individual cell rhythms give estimates of the order parameter R and phase Ω ; 2) scale R by β to account for 'experimental condition' (Ω is left unchanged); 3) assuming the population is desynchronized, invert Eq.(4) to find the PRC $g(\theta)$ or more specifically the parameters (a , α) in the underlying phase oscillator model that gives rise to the PRC. So that this method can be useful, it is imperative that the exposition of the method be improved greatly. It should be possible for a reasonably competent reader to implement the method for their own data. Here are some specific reasons why I find the method difficult to understand:

- a) The parameters ω and Δt in Eq.(4) are not defined until much later (after Eq.(12)) in a different context.
- b) Possibilities for the PRC $g(\theta)$ are given in Eq.(5) and Eq.(6), but those are for fitting to PRC data. The $g(\theta)$ resulting from the SR method is implicit in the phase oscillator model Eqs.(9) - (12). Perhaps give the expression for the PRC referenced in (17).
- c) All of the validation results of the method come before the method is adequately explained in the methods section. Perhaps the introduction could include a better summary of the details of the method or an example using synthetic data (generate 'data' from the phase oscillator model and then use the method to infer the known PRC).
- d) Use the index j instead of i to avoid confusion with the imaginary unit i .
- e) The SR method is better explained in Masuda et al. Nat. Commun. 2021. For completeness, the details there should be duplicated here.
- f) The details of the inversion process (step 3 above) are not described. This is a non-trivial root finding problem especially since the integral in Eq.(4) is a smoothing operation. I am surprised that the authors do not report experiencing any numerical difficulties (and how they were overcome) with this inversion.

In short, my primary suggestion is that the clarity of the description of the SR method be improved. The paper would also benefit from more investigation of the mechanism behind and the limits of the SR method. Why does it work? When does it not work?

Minor suggestions:

- 1) Change "mouse" to "mammalian" in the title since the approach was applied to multiple mammalian clocks.
- 2) line 26: "demonstrated" -> "demonstrate"; line 29: "PRC" -> "PRCs"; line 31: "revealed" -> "reveals" and "were" -> "are"; line 32: "revealed the" -> "reveal"
- 3) line 43: remove parentheses around "as a model of shift work"
- 4) line 49: remove "," after "in vivo"
- 5) line 81: "their" -> "its"
- 6) line 84: remove "the"
- 7) line 87: remove "the"
- 8) line 127: "Fig, 3b" -> "Fig. 1b"
- 9) "Forskolin" is inconsistently capitalized.
- 10) line 158: "also shows change" -> "also showed a change"
- 11) line 159: remove "also"
- 12) lines 163-165: This is a significant result. Explain in detail how Fig. 1b allows you to concluded that the SR method is effective for both damped and desynchronized rhythms.
- 13) line 188: remove "each other"
- 14) line 189: "of population" -> "of the population"
- 15) line 192: "as a SR" -> "as an SR"
- 16) line 215: remove "-" in " $6\text{-}\mu\text{m}$ " (this hyphen appears often between values and their units).
- 17) line 277: "became" -> "approached the" or "entered the"
- 18) line 278: "at" -> "before"
- 19) line 455: "because SCN" -> "because the SCN"
- 20) line 526: "point" -> "points"
- 21) line 549: "maximizing" -> "maximizing the"
- 22) Eq.(5),(6): Why use two rather than one or more harmonics?

23) line 555: "moving" -> "a moving"

24) Eq.(9): mention that $E(t)$ is a stimulus indicator

25) line 572: "PRC parameters" -> "SR parameters"?

26) line 569: How dependent is the SR method on this underlying phase oscillator model? Could you use the Stuart-Landau phase-amplitude model? 27) line 569: This phase oscillator model does not include coupling within the population. This is discussed in the context of the SCN in the manuscript. Could you include coupling in the SR method if you were willing to numerically simulate the PRC and solve for the model parameters entirely numerically?

Responses to reviewer's comments.

We thank all the reviewers for their constructive and valuable comments, which have helped us improve our manuscript significantly. We have carefully addressed each comment and revised the manuscript accordingly. The following are the authors' point-by-point responses to the reviewers' comments. We have earnestly tried to address all the concerns raised by the reviewers.

Reviewer #1 (Remarks to the Author):

This manuscript illustrates a new method for measuring circadian clock phase response curves (PRC), which was previously demonstrated in plants (the first author of the current manuscript is also the first author in the previous paper, [17] of the references). The method is based on the Singularity Response (SR) and it consists in applying a stimulus to a de-synchronized cell population. From the phase and amplitude of the population after stimulation, a mathematical model is then applied to recover the PRC parameters. In principle, the SR method is less costly and of more practical application than the classical PRC method, but it doesn't recover as many details.

The current work tests and illustrates this method for mammalian (mouse) cells, using several cell lines (NIH3T3, MEF, Rat1), as well as tissue slices; several different stimulus according to the cell line; and different clock reporters, also according to cell lines (Rev-erb, Per, Bmal1). The results are very interesting and comprehensive taking into account the range of cells, stimulus, and reporters tested. In a general way, the new method is in a fair-to-good correspondence with the more classical PRC methods, but when it comes to closer comparison of detailed properties of the SR and PCR, further work and studies are needed. Throughout the manuscript, some points could be further developed:

-In the validation for NIH3T3 cells, the authors perform tests with "small populations of 5 cells". It is not clear to me what is the goal of this test. It seems to me that 5 elements is too small for a population, I would expect groups of 20-30 cells in order to obtain reasonable averages. And how do you choose these "cell populations" (are they near neighbors?). On the other hand, how do "populations of 5 cells" compare with the population sizes in the other experiments, in other cell lines? Is it reasonable to assume that these small population results can be extrapolated to "real" populations?

Thank you for your insightful comment. In the original study, we created a small population containing five cells because we wanted to compare the phase response

between highly synchronized populations and desynchronized populations (original Figure 1e). When we create a large population, the synchronization rate should be close to the average synchronization rate of all cells, which is extremely low in this case. Thus, we intended to create relatively small populations. However, for the revised manuscript, we analyzed again the phase response in populations composed of 3, 5, 10, and 20 cells to confirm our hypothesis (New Supplementary Figure 4a). We concluded that the amplitude dependency of the phase responses at the cellular and population levels was also close to the mathematical model, as depicted in Fig. 1b.

We randomly selected cells and did not consider the spatial relationship among these cells when we created a small population (Fig. 1e, S4b). However, it is also possible to intentionally select cells in close phases to construct a highly synchronized population. In this case, PRCs are also close to the theoretical model (New Supplementary Figure 4b, Page 6, Line 161-165). A detailed method has been added to the relevant section of the revised manuscript.

- The results for Rat-1 cells are not always clear to understand and/or explain. First, the phases computed through SR seem to always be underestimated (all dots above the $y=x$ in Fig 4d). Is this a specific problem for these type of cells?

In the original study, the phase parameter of SR for melatonin was not consistent with that of PRC, while other stimulation showed a similar phase relationship. We repeated the comparison of SR and PRC parameters for melatonin with strictly managed experimental conditions (almost same passage numbers, same lot of culture medium etc.). We found that the response of melatonin was weak, leading to inconsistency in the phase of SR and PRC. As we described, the SR method is useful when the stimulation is strong. We also measured the PRC parameter for PMA again to confirm whether the underestimation of the SR phase is specific to Rat-1 cells. However, we obtained a similar result, in which the SR phase was slightly delayed compared to the PRC phase (Fig. 4d). It is possible that the SR phase tends to be underestimated, although the SR phase is almost compatible with the PRC phase.

Second, the PRC and SR parameters for Forskolin and PMA are very close to each other -- as indicated by the corresponding dots in Figs. 4c,d.

However, the reconstructed PRC curve for PMA fit the points very badly, in contrast to the Forskolin curve (Fig. 4e). Moreover, the PMA and Forskolin stimulation curves are

similar in amplitudes (both high after stimulation), which seems to indicate that PMA should also provide good results, as was indeed the case in NIH3T3 cells. Can these divergences be explained in some way?

We again measured the PRC for PMA with more strictly managed experimental conditions (almost same passage numbers and the same lot of culture medium as SR measurement, etc.). We found that predicted PRC using the SR method was similar to measured PRC. We speculated that the inconsistency in the two PRCs in the original study was caused by the small number of samples for measured PRC or differences in experimental conditions such as the passage numbers.

- The results for MEF cells do not include comparison with the classical PCR, but rather focus on the dependence of the phase response on the stimulus concentration, in particular for Dexamethasone (Fig. 5e,f). This is an interesting application of the SR method, where the SR phase and amplitude parameters are used to characterize phase response as a function of DEX concentration. Since the comparison is among SR parameters only, some meaningful information can be obtained (see also my next comment).

As per your suggestion, we have added the results of PRC measurements in MEFs treated with DEX (New Supplementary Fig. 12), confirming the reliability of the SR method, to the revised manuscript.

- Regarding the SR method itself, it can be shown that the SR phase parameter is equivalent to the phase parameter of the PCR method, i.e. $\theta(\text{PCR}) = \theta(\text{SR})$. This is, however, not true for the amplitude parameter, since a relation of the form $R(\text{PCR}) = \beta \cdot R(\text{SR})$ needs to be computed for each cell type and the factor β is different from cell line to cell line, and probably different for each experiment. Therefore, the interpretation of the amplitude parameter for SR can be a problem when recovering PCR and would require further study.

Nevertheless, in the case of comparison among SR experiments only (cf. the previous comment), the values $R(\text{SR})$ retain their meaning and can be used to compare and characterize results from similar experiments, such as the dependence of phase response on the concentration of the stimulus (see Fig. 5e,f). Indeed, as the authors suggest in the Discussion, the SR method could be more suitable for a preliminary, and faster, phase response analysis in experiments such as high throughput screening, where large amounts of tests need to be performed. The SR would enable a selection of the best cases, which would then be followed by a more detailed classical PCR.

As we described in the supplementary text, β can be estimated using the SR parameters

when the stimuli are strong enough to produce type 0 resetting (i.e. $R(\text{PRC})$ is 1). Herein, we confirmed that 100 nM DEX treatment actually resulted in the type 0 resetting using the conventional method (New Supplementary Fig. 12) and that beta estimated using only the SR parameter was accurate.

Other remarks:

Fig 1b,c: the captions need to be corrected, as the two panels include both single cell and population level(?)

We apologize for the confusion. We have edited the captions according to your comments.

Figs 2g and 4c: give a better idea of what the "approximate line" is, or at least refer to eq. (7) in the Methods.

We have edited the description.

Line 562 (Methods): Should be Fig 2g? (and not 2f)

We have corrected the manuscript as indicated.

Reviewer #2 (Remarks to the Author):

The authors apply their recently published (Masuda et al. Nat. Commun. 2021) singularity response (SR) method for inferring the phase response curve (PRC) from a single experiment to mammalian rather than plant circadian clocks. The method is successfully applied to experimental data in cultured mouse and rat fibroblast cells, as well as several mouse tissue slices for many different stimuli including temperature, PMA, DEX, Forskolin, sodium bicarbonate, LiCl, and CoCl₂. This variety of cell types and stimuli provide compelling evidence that the method is broadly useful. This is exciting since it promises to reveal the very insightful PRC with a single experiment from a desynchronized state rather than numerous experiments from a synchronized state for many different phases of the input.

After reading the methods section multiple times and consulting the previous paper on the SR method (Masuda et al. Nat. Commun. 2021), the SR method remains opaque. As I understand it, the method steps are 1) measurements of the individual cell rhythms give estimates of the order parameter R and phase Ω ; 2) scale R by β to account for 'experimental condition' (Ω is left unchanged); 3) assuming the population is desynchronized, invert Eq.(4) to find the PRC $g(\theta)$ or more specifically the parameters (a, α) in the underlying phase oscillator model that gives rise to the PRC. So that this

method can be useful, it is imperative that the exposition of the method be improved greatly. It should be possible for a reasonably competent reader to implement the method for their own data. Here are some specific reasons why I find the method difficult to understand:

a) The parameters ω and Δt in Eq.(4) are not defined until much later (after Eq.(12)) in a different context.

Based on your comment, we have added an explanation of the parameters in Eq. (4) in the section following Eq. (4). (Page 21, Line 592-593)

b) Possibilities for the PRC $g(\theta)$ are given in Eq.(5) and Eq.(6), but those are for fitting to PRC data. The $g(\theta)$ resulting from the SR method is implicit in the phase oscillator model Eqs.(9) - (12). Perhaps give the expression for the PRC referenced in (17).

We have added an alternative description of the experimentally obtained PRC $g(\theta)$ and $g(\theta)$ calculated using the simulation model as $g_{\text{exp}}(\phi)$ and $g_{\text{model}}(\phi, \Delta t)$, respectively.

c) All of the validation results of the method come before the method is adequately explained in the methods section. Perhaps the introduction could include a better summary of the details of the method or an example using synthetic data (generate 'data' from the phase oscillator model and then use the method to infer the known PRC).

As per your suggestion, we have added a summarized experimental method to the new Supplementary Figure 1 (Page 3-4, Line 68-96).

d) Use the index j instead of i to avoid confusion with the imaginary unit i .

We have corrected the text to avoid this confusion (Page 21, Line 594-600).

e) The SR method is better explained in Masuda et al. Nat. Commun. 2021. For completeness, the details there should be duplicated here.

We have added a description in the revised manuscript (Page 21-23, Line 587-644).

f) The details of the inversion process (step 3 above) are not described. This is a non-trivial root finding problem especially since the integral in Eq.(4) is a smoothing operation. I am surprised that the authors do not report experiencing any numerical difficulties (and how they were overcome) with this inversion.

We previously showed that when Δt is small enough compared to the period (at least approximately 1/3 of the period), R monotonically increases with an increase in a , as shown in Supplemental Figure 2a in Masuda et al., Nature Communication, 2021. Thus, we can uniquely determine a using R . We have added a detailed explanation to the supplemental text (Page 8, Line 214-218, Supplementary Fig. 8, Supplementary text).

In short, my primary suggestion is that the clarity of the description of the SR method be improved. The paper would also benefit from more investigation of the mechanism

behind and the limits of the SR method. Why does it work? When does it not work?

Thank you for your insightful comments, we have modified the manuscript thoroughly so that our method is easier to understand.

Minor suggestions:

1) Change "mouse" to "mammalian" in the title since the approach was applied to multiple mammalian clocks.

Thank you for your suggestion, we have edited the title.

2) line 26: "demonstrated" -> "demonstrate"; line 29: "PRC" -> "PRCs"; line 31: "revealed" -> "reveals" and "were" -> "are"; line 32: "revealed the" -> "reveal"

3) line 43: remove parentheses around "as a model of shift work"

4) line 49: remove "," after "in vivo"

5) line 81: "their" -> "its"

6) line 84: remove "the"

7) line 87: remove "the"

8) line 127: "Fig, 3b" -> "Fig. 1b"

9) "Forskolin" is inconsistently capitalized.

10) line 158: "also shows change" -> "also showed a change"

11) line 159: remove "also"

Thank you for your careful assessment. We have corrected the manuscript as per your suggestions in 2) to 11).

12) lines 163-165: This is a significant result. Explain in detail how Fig. 1b allows you to concluded that the SR method is effective for both damped and desynchronized rhythms.

In both models, the circadian clock is reset to a specific phase with a specific amplitude using stimulation in a singularity (damped or desynchronized) state. We have added a description in the relevant sections of the revised manuscript (Page 6. Line 168-169).

13) line 188: remove "each other"

14) line 189: "of population" -> "of the population"

15) line 192: "as a SR" -> "as an SR"

16) line 215: remove "-" in " $6\text{-}\mu\text{m}$ " (this hyphen appears often between values and their units).

17) line 277: "became" -> "approached the" or "entered the"

18) line 278: "at" -> "before"

19) line 455: "because SCN" -> "because the SCN"

20) line 526: "point" -> "points"

21) line 549: "maximizing" -> "maximizing the"

We have corrected the manuscript as per your suggestions 13) to 21).

22) Eq.(5),(6): Why use two rather than one or more harmonics?

PRCs were set to second-order harmonics because there is some noise in the experimentally measured PRCs and it is difficult to accurately determine the higher-order components of the PRC. We have added a description in the relevant section of the revised manuscript (Page 21. Line 597-599).

23) line 555: "moving" -> "a moving"

24) Eq.(9): mention that $E(t)$ is a stimulus indicator

25) line 572: "PRC parameters" -> "SR parameters"?

We have corrected the manuscript as per your suggestions 23) to 25).

26) line 569: How dependent is the SR method on this underlying phase oscillator model? Could you use the Stuart-Landau phase-amplitude model?

Although the parameters of SR and PRC correspond to each other regardless of the model used (Eq.(4)), the magnitude of PRC amplitude depends on the models used. However, because it is a general property of limit cycle oscillator models that type-1 PRC for weak stimulation approaches type-0 as stimulation becomes stronger, we expect that similar results can be obtained even in other models. The Stuart-Landau model can also be used to estimate PRC, but the PRC cannot be defined as a single form in this model when multiple parameters are present. Therefore, we used the simplest phase oscillator model. We have added a description in the relevant section of the revised manuscript (Page 22. Line 615-620).

27) line 569: This phase oscillator model does not include coupling within the population. This is discussed in the context of the SCN in the manuscript. Could you include coupling in the SR method if you were willing to numerically simulate the PRC and solve for the model parameters entirely numerically?

It is possible to perform numerical simulations considering coupling in the phase oscillator or the Stuart-Landau model. However, the coupling is not so important for PRC estimation, at least if we are considering instantaneous phase response. Even if the coupling causes a slight amplitude change, accurate SR parameters can be obtained upon correction of Eq. (7). If the coupling is strong enough to significantly affect the PRC estimation, it may be difficult for the cell population to reach the singularity state. However, spatially asymmetric couplings, such as those of cells in the SCN, may cause more complex phase changes, and more detailed analysis is needed for such cases. We have added a description to the relevant sections in the revised manuscript (Page 22. Line 625-628).

Other revision:

- 1) *It was difficult to distinguish between a and α in Eq. (10), so we corrected these to a and b .*
- 2) *We have corrected the letters of SR amplitude and phase and the SR parameters obtained from PRC to R'_{SR} , θ'_{SR} , R'_{PRC} and θ'_{PRC} , respectively.*
- 3) *Fig. 1h and Fig. 4 d were labeled opposite on the vertical and horizontal axes, which have now been corrected.*
- 5) *We have added more detailed description to the Methods section and Figure legends.*
- 6) *We have changed the order of Supplementary Figures.*
- 7) *We have improved figure readability.*

REVIEWERS' COMMENTS

Reviewer #1 (Remarks to the Author):

In their reply, the authors have answered all my questions and comments very suitably and convincingly. They have performed complementary experiments which confirmed and also improved the original results. The revised manuscript is improved and more clear. Therefore, I highly recommend this manuscript to be accepted.

Reviewer #2 (Remarks to the Author):

The explanation of the SR method is much better in this revision. I especially appreciated the description on lines 587-600 and the new figures in the supplement. The revised manuscript satisfactorily addresses the issues that I previously raised.

minor suggestions:

- 1) Eq. 4: Should g have a subscript "exp" as appears in the text after Eq. 4? If not, remove "Here, " from the paragraph following Eq. 4.
- 2) line 596: "Each PRCs are described as follows," -> "The PRCs have the assumed forms,"
- 3) Eq. S2: FYI I checked that this formula is correct.
- 4) Supp. Fig. 1: "to estimate PRC" -> "to estimate the PRC"; add "the" before PRC and SR in several other places.
- 5) Supp. Fig. 8: Mention that these are plots of the analytical expressions (S17) and (S18) from [17].
- 6) Eq. 7: end with a period.
- 7) Fig. 2de: insert a space between "Concentration" and "(μM)"
- 8) line 608: "proportional coefficient" -> "The proportionality coefficient β "; ", and may include" -> "accounts for"
- 9) line 217: "the phase of stable point of PRC" -> "the phase of the stable point of the PRC"
- 10) line 624: "in other" -> "otherwise"

Responses to reviewer's comments.

We thank all the reviewers for their constructive comments. We have revised the manuscript accordingly.

Reviewer #2 (Remarks to the Author):

The explanation of the SR method is much better in this revision. I especially appreciated the description on lines 587-600 and the new figures in the supplement. The revised manuscript satisfactorily addresses the issues that I previously raised.

minor suggestions:

1) Eq. 4: Should g have a subscript "exp" as appears in the text after Eq. 4? If not, remove "Here, " from the paragraph following Eq. 4.

We have added the subscript "exp" to g in Eq. 4.

2) line 596: "Each PRCs are described as follows," -> "The PRCs have the assumed forms,"

We have corrected it.

3) Eq. S2: FYI I checked that this formula is correct.

Thank you so much for confirming it.

4) Supp. Fig. 1: "to estimate PRC" -> "to estimate the PRC"; add "the" before PRC and SR in several other places.

We have corrected it.

5) Supp. Fig. 8: Mention that these are plots of the analytical expressions (S17) and (S18) from [17].

We have mentioned this in the figure legend.

6) Eq. 7: end with a period.

We have added a period.

7) Fig. 2de: insert a space between "Concentration" and "(μM)"

We have inserted a space.

8) line 608: "proportional coefficient" -> "The proportionality coefficient β "; ", and may include" -> "accounts for"

We have corrected it.

9) line 217: "the phase of stable point of PRC" -> "the phase of the stable point of the PRC"

We have corrected it.

10) line 624: "in other" -> "otherwise"

We have corrected it.